# First-Principles Study of Adsorption of CH_4_ on a Fluorinated Model NiF_2_ Surface

**DOI:** 10.3390/ma17092062

**Published:** 2024-04-27

**Authors:** Tilen Lindič, Beate Paulus

**Affiliations:** Institute for Chemistry and Biochemistry, Freie Universität Berlin, Arnimallee 22, 14195 Berlin, Germany

**Keywords:** periodic DFT, adsorption study, Simons process, fluorination of methane

## Abstract

Electrochemical fluorination on nickel anodes, also known as the Simons’ process, is an important fluorination method used on an industrial scale. Despite its success, the mechanism is still under debate. One of the proposed mechanisms involves higher valent nickel species formed on an anode acting as effective fluorinating agents. Here we report the first attempt to study fluorination by means of first principles investigation. We have identified a possible surface model from the simplest binary nickel fluoride (NiF_2_). A twice oxidized NiF_2_(F_2_) (001) surface exhibits higher valent nickel centers and a fluorination source that can be best characterized as an [F_2_]^−^ like unit, readily available to aid fluorination. We have studied the adsorption of CH_4_ and the co-adsorption of CH_4_ and HF on this surface by means of periodic density functional theory. By the adsorption of CH_4_, we found two main outcomes on the surface. Unreactive physisorption of CH_4_ and dissociative chemisorption resulting in the formation of CH_3_F and HF. The co-adsorption with the HF gave rise to four main outcomes, namely the formation of CH_3_F, CH_2_F_2_, CH_3_ radical, and also physisorbed CH_4_.

## 1. Introduction

Fluorinated compounds represent a large share of commodity chemicals and can be found in many pharmaceuticals [1,2] and agrochemicals [3]. Research into different fluorination methods is, therefore, attracting a lot of scientific interest. Considering the imperative for environmentally sustainable synthetic methods, electrochemistry can be an attractive solution [4]. A deeper understanding of the underlying processes would, therefore, help to optimize the (electrochemical) processes.

Electrochemical fluorination on nickel anodes (ECF) was introduced in the middle of the 20th century. The process is commonly known as the Simons process, after J. Simons, who first reported it in 1949 [5,6,7,8,9]. Shortly thereafter, the process was patented [10] and it has been used on an industrial scale since then. The setup of the Simons process is rather simple and consists of a nickel anode immersed in anhydrous hydrogen fluoride. The substrate is then added and the fluorination reaction proceeds readily after the application of external potential [11]. Despite the ease and the applicability of the Simons process, its mechanism is not completely understood and is under continuous debate. Two possible mechanisms have been discussed in the literature. The first, known as the EC_b_EC_N_ mechanism, involves the initial formation of a radical cation via direct electrochemical oxidation of the substrate (E). This is followed by a deprotonation step that generates a radical (C_b_), which then undergoes electrochemical oxidation (second E step) to form a carbocationic intermediate, to which in the last step the F^−^ is added (final C_N_ step) [12]. Although this mechanism accounts for some of the products observed in the Simons process, there is compelling evidence against it. Specifically, the fact that many experiments were reported where the Simons cell was pre-electrolyzed and the successful fluorination of the substrate was observed even when it was added to the cell after the external potential was discontinued [13,14].

The second mechanism discussed in the literature involves higher valent nickel fluorides formed on the anode acting as a fluorination source [15]. Many experimental studies provided indirect evidence corroborating this mechanism [16,17,18,19,20]. It is proposed to proceed in two steps: (I) formation of the Ni_*x*_F_*y*_ film on the anode under an external potential and (II) fluorination of the substrate by the Ni_*x*_F_*y*_ film formed on the anode surface. Another strong piece of evidence for this mechanism is the aforementioned fact that fluorination can readily proceed even after the electric potential is discontinued. It was proposed that the Ni_*x*_F_*y*_ film consists of NiF_3_ or NiF_4_ [11]. This is substantiated by the fact that it was shown that NiF_3_ can be a strong fluorinating agent [21]. More recently, a combined experimental and theoretical study showed by means of cyclic voltammetry, that the surface nickel atoms are oxidized from +2 to +3 at around the operational potential of the Simons process [22]. Further corroborating the presence of higher valent nickel centers is the recent in-situ XANES study, which, for the first time, confirmed the existence of high valent nickel centers on the anode at high potentials [23].

Simons-type electrochemical fluorination of methane was first reported by Sartori [24]. He predominantly observed the formation of CF_4_ and CHF_3_ and only small quantities of CH_2_F_2_ and CH_3_F. Further work was conducted by Nagase et al. [25]. They developed a way to obtain different ratios of products by changing the parameters of the reaction leading to predominantly obtaining partially fluorinated products.

In this work, we study the reaction of methane on a twice oxidized NiF_2_ surface by first principles. This surface was not yet reported in the literature but was identified in a previous study in our group [22]. In addition to exhibiting higher valent nickel surface atoms, it also possesses an [F_2_]^−^ unit which is readily available to fluorinate the adsorbate. Furthermore, the surface shows enhanced stability under the potentials close to those utilized in the Simons process, making it a good candidate as a model surface for electrochemical fluorination (for the plot of stability of the investigated surface against other surface cuts see Appendix A).

## 2. Computational Details

All the systems were studied via periodic, spin unrestricted density functional theory (DFT) as implemented in Vienna ab-initio simulation package (VASP) version 5.4.4 [26,27,28]. The Perdew–Burke–Ernzerhof (PBE) [29] exchange-correlation functional was used in all calculations as was shown in a previous study to describe the system under investigation [22]. Plane waves combined with projector augmented wave (PAW) potentials [30,31] were used as a basis set. Within this method, the 2s and 2p orbitals of fluorine and the 3d and 4s orbitals of nickel were explicitly treated, with all other orbitals being implicitly accounted for within the core. Plane waves with a kinetic energy of up to 700 eV were included. Hubbard correction in the framework of the Dudarev approach [32] was added on top of the PBE functional for the strongly correlated Ni d electrons. The effective U value used was 5.3 eV [22]. To account for long-range interactions, Grimme’s D3 dispersion correction [33] with Becke–Johnson damping [34] was added on top of the PBE functional. The first Brillouin zone was sampled via an automatically generated, Γ centered Monkhorst–Pack K-point grid with the 8 × 8 × 1 mesh for the surface calculations. Gaussian smearing with the smearing width of 0.10 eV was used in all calculations. Electronic energy minimization was performed with the blocked Davidson algorithm with a convergence criterion of 10^−6^ eV for structural relaxation and single point calculations, respectively, and 10^−8^ eV for the calculation of vibrational frequencies. The structural relaxation was performed with the conjugate gradient algorithm with the convergence criterion for forces smaller than 0.01 eV/Å. In the structural relaxation, only the ionic positions were allowed to relax, with the cell volume and shape being fixed. Bader charges were calculated with the external code developed by the Henkelman group [35]. The molecules were calculated in a three-dimensional box of 15 Å in length. Vibrational frequencies were calculated using a finite differences approach with the step width of 0.02 Å. To find possible transition states, a climbing image nudged elastic band (CI-NEB) was employed [36,37]. Four images between the initial and the final states were calculated. VESTA was used for the visualization of the structures [38].

### Structural Model

The surface model is shown in Figure 1a. A stoichiometric cut of the NiF_2_ (001) surface does not have the [F_2_]^−^ unit; therefore, this surface can be described as being twice oxidized (for the simplicity of notation we label it as (NiF_2_)F_2_ (001) surface). Formally, the Ni centers in the stoichiometric NiF_2_ (001) surface are in oxidation state +2. The magnitude of their calculated magnetic moment is 1.830 μ_B_ which corresponds to a Ni(II) d^8^ electron configuration with two unpaired electrons. The Ni atoms on the surface, in a +4 formal oxidation state, exhibit a magnetic moment of 2.218 μ_B_. This suggests a deviation from the expected d^6^ Ni(IV) electron configuration. Instead, it points towards a high spin d^7^ Ni(III) with three unpaired electrons in the d shell. This conclusion is drawn under the assumption of a perfect octahedral environment and simplified splitting of the d orbitals in an octahedral field. This conclusion is further substantiated by the magnetism exhibited on the surface of fluorine atoms. The magnitude of the magnetic moment on each of them is 0.595 which can correspond to roughly half an electron. It can, thus, be concluded that there is one electron associated with the [F_2_] unit on the surface; therefore, we label it as an [F_2_]^−^ unit. The Ni–F distances in the bulk region are 2.014 Å and 2.015 Å, whereas both the surface Ni–F distances are significantly shorter at 1.881 Å. The distance between the surface fluorine atoms is 1.997 Å.

Six different high symmetry positions for the adsorption of CH_4_ were identified on the surface (see Figure 1c). Site 0 can be classified as on top of the fluorine atom, sites 1, 2, 3, and 4 as the bridge, and site 5 as hollow. For the adsorption of only CH_4_, the molecule was placed at 1.5, 1.7 and 2.0 Å away from the surface (the distance between the topmost fluorine of the surface and the bottommost atom of the adsorbate), respectively, on each of these positions. For the co-adsorption with the HF, the combinations of all possible positions were made. This resulted in 30 different starting structures. Furthermore, three different orientations of HF were considered (see Figure 1d): F atom pointing towards the surface (labeled as down), F atom pointing away from the surface (labeled as up) and HF molecule being parallel to the surface (flat). For the co-adsorption, the molecules were placed 1.7 Å from the surface. The starting configurations are labeled with numbers, for example, 0_1 represents methane adsorbed on position 0 and HF adsorbed on position 1. In all cases, the molecules were adsorbed on both the top and the bottom side of the surface. In the case of adsorbed CH_4_, the whole slab model was allowed to relax. For the co-adsorption studies only for the case of down orientation of HF the whole slab was allowed to relax. Since we found that the inner layers of the slab remained unchanged and in light of computational effort, only the three top- and bottommost layers were allowed to relax for the flat and up orientations of HF.

The adsorption energy was calculated as
(1)$$E_{\mathrm{ads}}=E_{\mathrm{surface}/\mathrm{ads}}-\left(E_{\mathrm{surface}}+E_{\mathrm{adsorbate}}\right),$$
where the *E*_surface/ads_ is the energy of the surface and adsorbates after the relaxation, *E*_surface_ is the energy of the clean NiF_2_(F_2_) surface and *E*_adsorbate_ is the energy of all the adsorbates on the surface calculated as free molecules. The charge transfer was calculated via the Bader charges by: (2)$$\Delta q=q_{i,\;\mathrm{free}}-q_{i,\;\mathrm{surface}/\mathrm{ads}},$$
where *q*_i, free_ is the Bader charge of the isolated species i (for the surface atoms that are the clean surface and for the molecule, the calculated Bader charges in vacuum) and *q*_i, surface/ads_ is the Bader charge of species i in the adsorbed system. By this definition, the positive values of Δq represent the gain of electron density and the negative values depletion of electron density.

## 3. Results and Discussion

### 3.1. Adsorption of CH_4_

For the adsorption of CH_4_ on the twice oxidized (NiF_2_)F_2_ (001) surface, 18 starting structures were considered, with two of them not showing convergence and were, therefore, discarded. After the structural optimization, two main outcomes were observed: (1) barrierless formation of CH_3_F and (2) a weak physisorption of CH_4_. Despite the availability of another fluorine atom and the theoretically possible formation of CH_2_F_2_, this was not observed. This can be explained by the probable high energy barrier of the cleavage of the second C–H bond. Within the formation of CH_3_F, there are three distinct cases (See Figure 2): dissociative adsorption via cleavage of the C–H bond and adsorption of CH_3_ through one of the surface fluorine atoms, while the hydrogen migrates to the other fluorine atom and forms adsorbed HF. This outcome was observed for three different structures, resulting in slightly different adsorption modes. The energetically most stable of them had an adsorption energy of −10.28 eV, which was also overall energetically the most favorable outcome (see Table 1). The second lowest in energy (at −9.30 eV) is the case where both CH_3_F and HF are desorbed from the surface, which happened only for one starting structure. Following are the adsorption modes where CH_3_F is desorbed but the hydrogen is not adsorbed on the surface fluorine atom, but rather on the fluorine in the plane of surface nickel atoms. This happened in six cases with all of the adsorption energies being within the same order of magnitude, from −8.59 eV to −8.45 eV.

Methane is weakly physisorbed in six structures, with the adsorption energies ranging from −0.21 eV to −0.17 eV. All the minima were confirmed by the calculation of vibrational frequencies, which did not show any significant imaginary values.

Magnetic moments and charge transfer of the surface nickel and surface fluorine atoms for all four cases are collected in Table 1. The trends for the other structures of the same type are similar and the data can be found in the Appendix A. From the magnetic moments on the Ni atom in structures (a) to (c) it can be concluded that previously Ni(III/IV) centers were reduced to Ni(II), with characteristic magnetic moments of around 1.8 $\mu_{B}$, roughly the same value which is also observed in the bulk regions of the slab. This is also corroborated by the values of the charge transfer for Ni atoms, with a value of around 0.3 e. The positive value indicates a gain in electron density. Magnetic moments on the surface fluorine atoms are almost 0 $\mu_{B}$ indicating a closed shell F^−^. This also explains significant electron density gain as indicated by a large positive charge transfer. Charge transfer on the carbon atom is highly negative, which can be explained by the fact that the hydrogen atom is substituted by fluorine which is much more electronegative, and therefore, takes the electron density away from the carbon. For the type of structures where CH_4_ is only weakly physisorbed on the surface, it can be seen that there is almost no change in the magnetic moments of nickel and fluorine, and there is no significant charge transfer on any of the species. This also explains very small adsorption energy and indicates that there is almost no interaction of the adsorbate with the surface.

Some selected structural parameters for the structures (a)–(d) are collected in Table 2. Ni–F distances in structure (a) are significantly elongated compared to the clean surface (1.881 Å), with the longer one (2.156 Å) corresponding to the fluorine through which the methane is adsorbed. This is not surprising since the surface Ni atoms are in oxidation state +2 and the distances are even larger as those around the Ni(II) centers in the bulk (2.014 Å) since the fluorine is bound to both nickel and carbon. In structure (b) where CH_3_F is desorbed, the shortest Ni–F distance between the surface Ni and the fluorine of HF is significantly longer. In structures (c) and (d) the Ni–F distances are comparable to those on the clean surface (1.881 Å). The C–F bond lengths are slightly longer compared to the free molecule (1.405 Å), which is a consequence of the interaction with the surface. The H–F distances clearly show two different situations, namely the desorbed HF molecule in (b) where the bond length is almost the same as in the free molecule (0.938 Å) and the adsorbed hydrogen in (a) and (c), where the bond length is slightly elongated.

### 3.2. Co-adsorption of CH_4_ and HF

Because the motivation for this project is the Simons process, which occurs in anhydrous HF, the next step after the adsorption of CH_4_ was the study of the co-adsorption of CH_4_ and HF. As already described, three different orientations of the HF molecules were considered. In total 90 different starting structures were generated (Note, that not all the structures converged and those were, therefore, excluded). All the resulting structures and their corresponding data are collected in the Appendix A.

Following the geometry relaxation the resulting structures were categorized into four distinct groups, reflecting their structural resemblances. Each group was further subdivided into subgroups. The lowest energy of each subgroup structure is shown in Figure 3. Group I represents the mono-fluorination of CH_4_ and the formation of CH_3_F. In group II the CH_4_ is doubly fluorinated yielding CH_2_F_2_ and H_2_. In group III there is no fluorination, but the CH_3_ radical-like species is formed. The last group represents weakly physisorbed CH_4_ and HF with no chemical change to the adsorbates. When taking into account only the most stable structures of each group (and their corresponding subgroups), it can be clearly seen that the adsorption energies of different groups fall into different energy ranges.

#### 3.2.1. Group I

The first group corresponds to the mono-fluorination of CH_4_ and the formation of CH_3_F. It can be further subdivided into four groups depending on what happens to the co-adsorbed HF (see Figure 4). In total, there are 50 structures in this group.

Figure 4a, with the lowest adsorption energy of −1.53 eV corresponding to physisorbed CH_3_F and two adsorbed HF molecules. In this case, the source of the fluorine atom in CH_3_F is the co-adsorbed HF molecule. The remaining hydrogen is adsorbed via one of the surface fluorine atoms. The hydrogen which comes from the methane is then adsorbed to the other surface fluorine. Figure 4b represents the breaking of one of the methane C–H bonds and chemisorption through one of the surface fluorine atoms. The hydrogen migrates to the other surface fluorine, whereas the HF molecule remains physisorbed. The lowest adsorption energy in this subgroup is −1.28 eV. Figure 4c is similar to the Figure 4b, with the exception that desorption of the surface fluorine atoms occurs, resulting in the physisorbed CH_3_F and HF. The adsorption energy of the most stable structure is similar to group (b) at −1.15 eV. In the last Figure 4d the methane hydrogen is adsorbed to the fluorine atom belonging to the same layer as the nickel atoms. In these cases, the surface fluorine is not hydrogenated. The lowest adsorption energy in this group is −1.10 eV.

Magnetic moments for the subgroups of group I are shown in Table 3. In all the structures the magnetic moment on the surface Ni atom is around 1.8 $\mu_{B}$, which is indicative of Ni(II). Positive charge transfer also indicates a gain of electron density and, therefore, signifies the transition from a higher to a lower oxidation state. Surface fluorine atoms do not exhibit any significant magnetic moments, but show a highly positive charge transfer, which leads to conclude that they have gained a significant amount of electron density. Negative values of charge transfer on the carbon indicate that the electron density was lost. This can be explained by the structure since hydrogen was substituted with fluorine, which is much more electronegative (cf. structure (a) for the adsorption of CH_4_). There was no significant charge transfer on the fluorine originating from co-adsorbed HF. The reason for this is that in all four cases, the fluorine is in a similar environment as in the free HF (i.e., bound to hydrogen).

As expected for Ni(II) centers, Ni–F bond lengths are around 2 Å (see Table 4). The longest are in structure (c), where all the molecules appear to be desorbed from the surface. An exception is structure (d), where hydrogen is not adsorbed through the surface fluorine atom, but through the fluorine in the plane of Ni atoms. In this case, the Ni–F bond distance is 1.873 Å which is very similar to the clean surface. All the H–F bond distances are only slightly distorted from the value for isolated molecules.

#### 3.2.2. Group II

The second group corresponds to the formation of difluoromethane on the surface. Energetically this is the second most stable outcome. In total, there are four structures in this group which can be divided into two subgroups (see Figure 5). In all cases, the HF molecule splits and the fluorine attaches to methane, while the hydrogen atoms form an H_2_ molecule. The second hydrogen in CH_4_ is substituted by one of the surface fluorine atoms. The hydrogen then either attaches to the remaining surface fluorine atom (subgroup a) or migrates to the fluorine which is in the plane of nickel atoms (subgroup b). The most stable structure in the subgroup (a) has an adsorption energy of −1.12 eV. The structure in group (b) is significantly higher in energy at −1.06 eV.

Magnetic moments and charge transfer for selected atoms for group II structures are shown in Table 5. As can be expected from the structure (fluorination of methane through one and hydrogen abstraction through the other surface fluorine atom) magnetic moment on the surface nickel indicates Ni(II). This is also indicated by the positive value of charge transfer. Both the surface fluorine atoms also show significant positive charge transfer. Charge transfer on carbon atoms is highly negative. This is not surprising, since in both structures two hydrogens are substituted with fluorine atoms.

Selected structural parameters for group II are collected in Table 6. As expected for the Ni(II) centers, the Ni–F bond lengths are around 2.0 Å. Similar to the observation in structure I-(d), in structure II-(b) one Ni–F bond distance is much shorter. This is attributed to the absence of adsorption through fluorine atoms by either methane or hydrogen. One of the C–F bonds is slightly shorter than the other. Compared to the free molecule, where C–F bond distances are 1.378 Å, the distortion can be attributed to the interaction with the surface.

#### 3.2.3. Group III

In group III there are five structures. From methane, a radical like CH_3_ species is formed. Both the surface fluorine atoms are still coordinated to the Ni center. Hydrogen is adsorbed through one of them, while the other one remains free. Interestingly, the physisorbed HF is not the same as initially co-adsorbed HF, but rather a combination of hydrogen originating from CH_4_ and fluorine originating from the HF. The adsorption energy of the most stable structure is −1.26 eV.

Selected magnetic moments and charge transfer values are collected in Table 5. It can be seen that the magnetic moment on nickel atoms stays roughly the same as on the clean surface, indicating a high valent nickel center. The free fluorine atom on the surface has a magnetic moment of −1.237 $\mu_{B}$, whereas the one with the adsorbed hydrogen does not show any significant magnetic moment. The same trend is seen for the charge transfer, which is almost 0 e for nickel and positive for both the fluorine atoms, but with different magnitudes. Charge transfer on the fluorine originating from HF is almost 0, which can be attributed to the bond breaking the fluorine atom being in the same environment as in the free molecule. The magnetic moment on the carbon atom is 0.264 $\mu_{B}$, which is similar to the free CH_3_radical (0.298 $\mu_{B}$). There is a slight positive charge transfer (0.153 e). This points to a radical character of this species.

Selected structural parameters for group III are shown in Table 7. It can be seen that one of the Ni–F distances is significantly shorter than the other. This is similar to the other structure where one of the surface fluorine atoms is free and the other one facilitates the adsorption of hydrogen atoms. Two C–H bond distances are 1.089 Å and the third one is 1.092 Å. In free CH_3_ all the distances are 1.086 Å. The angles of the CH_3_ radical formed are 118.86°, 118.71° and 119.69°, respectively. This is very close to the perfect 120° in the planar CH_3_ radical, further pointing to the radical-like character of the formed species.

#### 3.2.4. Group IV

The last group includes the structures where methane does not directly react with the surface and remains physisorbed. In total, there are 26 such structures. In all but one structure, the HF molecules remain physisorbed. In one case (labeled as subgroup (b), see Figure 5) the HF molecule is split and the hydrogen migrates to one of the surface fluorine atoms, whereas the fluorine sits between the other surface fluorine atom and the adsorbed hydrogen. The adsorption energies of this group range from −1.42 eV to −1.18 eV. The structure in the subgroup (b) is the second lowest in energy at −1.38 eV.

As expected the magnetic moments on surface nickel centers for group IV structures are around 2.2 $\mu_{B}$ indicating that the higher valent nickel is present. This can also be seen for charge transfer which is almost zero. In both structures (a) and (b) there seems to be some interaction of the surface fluorine atoms with the adsorbates, which is indicated by changes in magnetic moments (and some charge transfer). There is no charge transfer on the carbon atom, which is not surprising since there is no change in the structure. The fluorine atom from the HF has a magnetic moment of −1.397 $\mu_{B}$ in structure (a) and 0.512 $\mu_{B}$ in structure (b). In both cases, the value of charge transfer is negative indicating loss of an electron. This is because the hydrogen is abstracted by the surface fluorine atom and the previously bound fluorine is almost a fluorine anion. This can also be seen by an elongated H–F bond length. In other structures of group IV where the HF molecule is not so close to the surface this effect is much smaller or not present.

As expected for this structure type (where both surface fluorine atoms remain coordinated to the nickel), the Ni–F bond lengths remain very similar to that of the clean surface. A slight elongation is seen for one closer to the hydrogen of HF (see Table 7).

#### 3.2.5. NEB Calculation

For the calculation of possible transition states for the transformation between adsorbed structures, we focused on the transformation between groups I and II. These two represent energetically the lowest adsorption modes and correspond to the second fluorination step of methane, leading from CH_3_F to CH_2_F_2_. Furthermore, this transformation is particularly interesting to explore because it can further illuminate how the fluorination itself may proceed. To achieve this, we have employed the CI-NEB calculations.

The energy profile of the transformation is shown in Figure 6. The calculated activation barrier of 10.18 eV is both thermodynamically and kinetically unfavorable. The vibrational frequencies of the transition state (image 4) showed one imaginary frequency at 472.9 cm^−1^ (for the depiction of the displacements associated with this imaginary frequency see Appendix A), which characterizes this structure as a true transition state (There is another imaginary frequency at 58.4 cm^−1^, but because it is significantly lower it can be ignored).

The structures of the calculated images are shown in Figure 7. The initial state is the group I structure of co-adsorbed CH_4_ and HF. The transition to the first image involves rotation of the physisorbed CH_3_F. Further rotation of the adsorbed HF happens in the next step. In the third image, the H_2_ molecule is formed from two hydrogen atoms of CH_3_F. This step represents a significant jump in the energy, which can be attributed to the formation of H_2_. In the fourth image, which is overall the highest in energy, and therefore, the transition state, the hydrogen atom adsorbed through the surface fluorine migrates to the fluorine in plane with nickel. A CHF fragment is adsorbed through the freed fluorine atom. This arrangement then leads to direct relaxation into the final state, where the hydrogen migrates back to the carbon, and the desorption of CH_2_F_2_ takes place.

A high energy barrier might indicate that, at least within this model system, the fluorination does not occur stepwise as could be expected, but rather the paths leading to different fluorinated products are independent of each other. This could also explain why in the real Simons process all the different degrees of fluorinated products are observed. It should be kept in mind that in the real Simons process, next to the fluorine source on the surface, there is also more fluorine available from the anhydrous HF. Furthermore, the formed hydrogen could be dissolved in the liquid HF thus lowering the barrier and avoiding the formation of H_2_ molecule.

## 4. Conclusions

The adsorption of methane and the co-adsorption of methane and hydrogen fluoride on a twice oxidized NiF_2_(F_2_) surface were investigated. The surface was identified as a possible model surface to describe the fluorination reactions as they happen in the Simons process. In the case of the adsorption of CH_4_, four distinct structural types were identified on the surface. The dissociative chemisorption of CH_4_ occurs through the surface fluorine atoms and the migration of hydrogen atoms either to one of the surface fluorine atoms or to the fluorine atom in plane with the surface nickel. Fluorination of CH_4_ and formation of CH_3_F and HF physisorbed on the surface or weak physisorption of CH_4_.

The results of the co-adsorption of CH_4_ and HF can be grouped into four different structural motives as follows: Monofluorination of CH_4_, difluorination of CH_4_, formation of CH_3_ radical and a weak physisorption of CH_4_. A single transition state was located for the transformation from mono- to difluorinated methane.

In the future, more hydrogen fluorides can be added to the surface, which would simulate the reality of anhydrous HF in the Simons process even closer. This would undoubtedly lead also to higher degrees of fluorination of methane which are also observed in the experiments. It is important to mention the limitations of the model. As the catalytically active nickel fluoride film is amorphous, there is no direct experimental proof of the structure of the active species. But our selected model, in its simplicity, is the first structural attempt to theoretically describe the fluorination reaction in the Simons process which takes into account the experimental observations known up to now.

## Figures and Tables

**Figure 1 materials-17-02062-f001:**
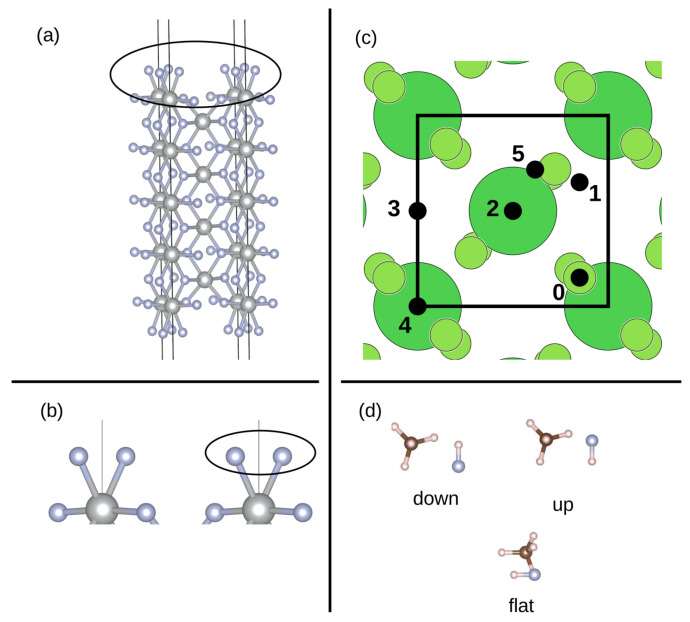
(**a**) Structure of the full (NiF_2_)F_2_ (001) surface model including vacuum, with the black circle labeling the top Ni layer and the [F_2_]^−^ unit. Ni atoms are shown in gray and F atoms in blue. (**b**) Zoom on the top of the surface with the [F_2_]^−^ unit labeled in black circle. Ni atoms are shown in gray and F atoms in blue. (**c**) Top view of the (NiF_2_)F_2_ (001) surface, black circles representing the investigated adsorption sites, with their corresponding numbering next to them. Large green circles represent Ni atoms, small green circles represent F atoms. (**d**) Three possible starting orientations of the co-adsorbed HF molecule: F pointing towards surface (down), F pointing away from surface (up) and HF molecule in parallel to the surface (flat). F atoms are in blue, C atoms in brown and H atoms in white.

**Figure 2 materials-17-02062-f002:**
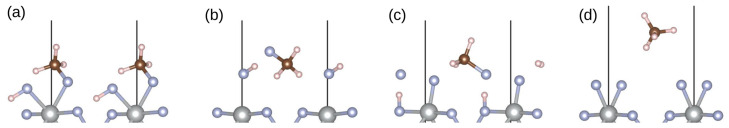
Structures of different outcomes of adsorption of CH_4_ on NiF_2_(F_2_) (001) surface. (**a**) formation of CH_3_F, chemisorbed on the surface; (**b**) formation of CH_3_F and HF, both physisorbed; (**c**) formation of CH_3_F and chemisorption of H; (**d**) physisorption of CH_4_. Ni atoms are represented in gray, F atoms in blue, C atoms in brown and H atoms in white colour.

**Figure 3 materials-17-02062-f003:**
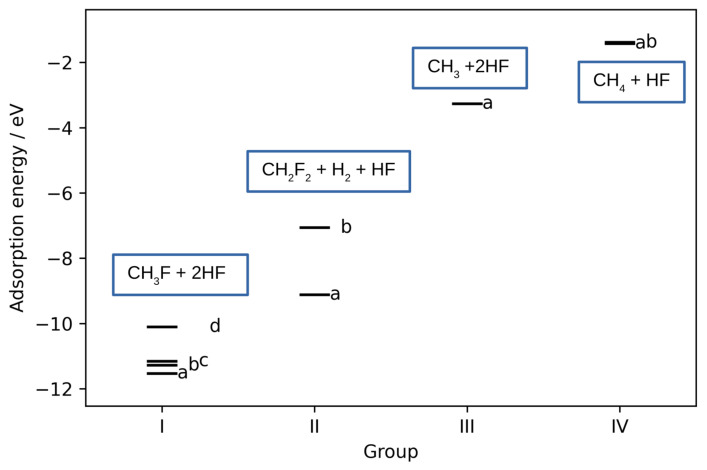
Adsorption energies of the most stable structures obtained after co-adsorption of CH_4_ and HF. Structures are divided into 4 groups based on their geometric similarities. Each group is further subdivided into subgroups denoted by letters. The main outcome for each of the group is written in the blue box.

**Figure 4 materials-17-02062-f004:**
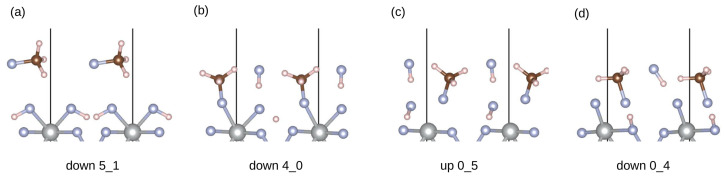
Structures in group I with the lowest adsorption energies are divided into four different subgroups (**a**–**d**). The initial orientation for each structure is written below. Ni atoms are represented in gray, F atoms in blue, C atoms in brown and H atoms in white color.

**Figure 5 materials-17-02062-f005:**
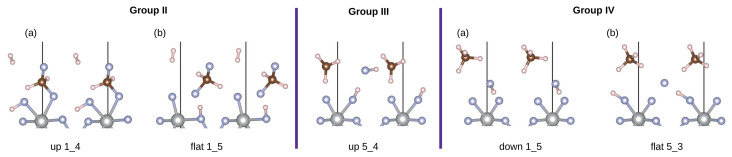
Structures in groups II, III and IV, respectively. Subgroups are denoted with letters. The initial orientation for each structure is written below. Ni atoms are represented in gray, F atoms in blue, C atoms in brown and H atoms in white color.

**Figure 6 materials-17-02062-f006:**
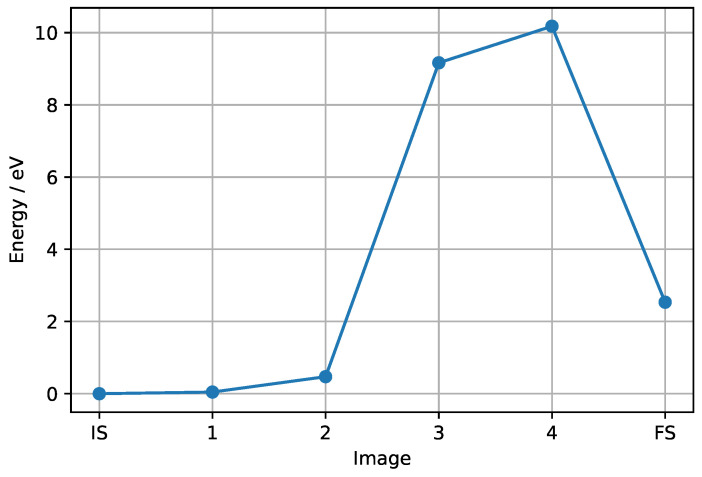
Energy profile of the NEB calculation for the transformation of group I (starting structure with down orientation of HF, position 5_1) co-adsorbed CH_4_ and HF to group II (starting structure with down orientation of HF, position 2_1).

**Figure 7 materials-17-02062-f007:**
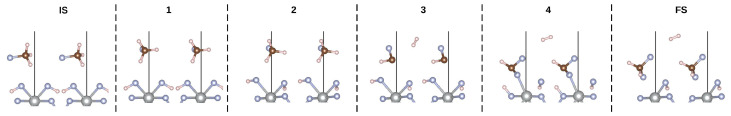
Structures of the initial state (IS), the final state (FS) and the four calculated images in between. Ni atoms are represented in gray, F atoms in blue, C atoms in brown and H atoms in white colour.

**Table 1 materials-17-02062-t001:** Adsorption energies (E*_ads_* in eV), magnetic moments (μ in $\mu_{BM}$) for Ni and surface F atoms and charge transfer (Δq in e) for Ni, surface F atoms and C for structures (a)–(d) (see Figure 2).

Structure	E*_ads_*	μ (Ni)	μ (F)	μ (F)	Δq (Ni)	Δq (F)	Δq (F)	Δq (C)
(a)	−10.28	−1.827	−0.014	−0.022	0.279	0.349	0.494	−0.534
(b)	−9.30	−1.819	−0.008	−0.000	0.294	0.491	0.378	−0.496
(c)	−8.59	−1.818	−0.060	−0.000	0.314	0.449	0.371	−0.370
(d)	−0.21	−2.218	−0.582	−0.590	0.000	0.010	0.014	−0.040

**Table 2 materials-17-02062-t002:** Some selected bond distances (in Å) for structures (a)–(d) (see Figure 2). NiF represents the distance between surface nickel and fluorine atoms, CF is the distance between carbon and fluorine and HF the distance between hydrogen and fluorine.

Structure	d (NiF)	d (NiF)	d (CF)	d (HF)
(a)	2.067	2.156	1.433	1.047
(b)	2.275	-	1.456	0.984
(c)	1.828	-	1.467	1.030
(d)	1.879	1.880	-	-

**Table 3 materials-17-02062-t003:** Adsorption energies (E*_ads_* in eV), magnetic moments (μ in $\mu_{BM}$) for Ni and surface F atoms and charge transfer (Δq in e) for Ni, surface F atoms, C and F of co-adsorbed HF (denoted as F_H_ for group I structures (a)–(d) (see Figure 4).

Structure	$E_{\mathrm{ads}}$	μ (Ni)	μ (F)	μ (F)	Δq (Ni)	Δq (F)	Δq (F)	Δq (C)	Δq (F*_H_*)
(a)	−11.53	−1.837	−0.023	−0.022	0.273	0.448	0.465	−0.548	−0.141
(b)	−11.28	−1.830	−0.020	−0.014	0.269	0.467	0.365	−0.445	0.000
(c)	−11.15	−1.833	−0.013	−0.016	0.270	0.351	0.493	−0.463	0.022
(d)	−10.10	−1.830	−0.008	−0.037	0.284	0.357	0.478	−0.419	0.012

**Table 4 materials-17-02062-t004:** Some selected bond distances (in Å) for group I structures (a)–(d) (see Figure 4). NiF represents the distance between surface nickel and fluorine atoms, CF is the distance between carbon and fluorine and HF the distance between hydrogen and fluorine.

Structure	d (NiF)	d (NiF)	d (CF)	d (HF)	d (HF)
(a)	2.030	2.033	1.413	1.013	1.013
(b)	2.068	2.110	1.452	1.118	0.965
(c)	2.212	2.246	1.452	1.088	0.961
(d)	1.873	2.455	1.490	1.011	0.996

**Table 5 materials-17-02062-t005:** Adsorption energies (E*_ads_* in eV), magnetic moments (μ in $\mu_{BM}$) for Ni and surface F atoms and charge transfer (Δq in e) for Ni, surface F atoms, C and F of co-adsorbed HF (denoted as F_H_) for group II, III and IV structures (see Figure 5).

Structure	$E_{\mathrm{ads}}$	μ (Ni)	μ (F)	μ (F)	Δq (Ni)	Δq (F)	Δq (F)	Δq (C)	Δq (F*_H_*)
II-(a)	−9.12	−1.827	−0.012	−0.022	0.282	0.343	0.494	−1.099	−0.157
II-(b)	−7.09	−1.816	−0.002	−0.059	0.319	0.347	0.465	−1.031	−0.148
III	−3.26	−2.227	−0.039	−0.237	0.002	0.439	0.323	0.153	−0.046
IV-(a)	−1.42	−2.240	−0.147	−0.581	−0.015	0.353	0.030	−0.002	−0.356
IV-(b)	−0.76	−2.224	0.181	−0.065	−0.011	0.067	0.433	−0.008	−0.467

**Table 6 materials-17-02062-t006:** Some selected bond distances (in Å) for group II structures (a) and (b) (see Figure 5). NiF represents the distance between surface nickel and fluorine atoms, CF is the distance between carbon and fluorine, HF the distance between hydrogen and fluorine and HH the distance between hydrogen atoms.

Structure	d (NiF)	d (NiF)	d (CF)	d (CF)	d (HF)	d (HH)
II-(a)	2.063	2.194	1.369	1.421	1.047	0.752
II-(b)	1.830	2.860	1.357	1.452	1.010	0.752

**Table 7 materials-17-02062-t007:** Some selected bond distances (in Å) for group III and IV (see Figure 5). NiF represents the distance between surface nickel and fluorine atoms, CF is the distance between carbon and fluorine, HF the distance between hydrogen and fluorine.

Structure	d (NiF)	d (NiF)	d (HF)	d (HF)
III	1.775	2.063	0.976	0.994
ine IV-(a)	1.837	1.930	1.076	-
IV-(b)	1.836	1.945	1.058	-

## Data Availability

Data are available from Appendix A. Further data are available from the authors on a reasonable request.

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
