# Peer review of "First-Principles Study of Adsorption of CH4 on a Fluorinated Model NiF2 Surface"

_materials, 2024, doi:10.3390/ma17092062_

Round 1

Reviewer 1 Report

Comments and Suggestions for Authors

The manuscript systemically studies the mechanism of fluorination reaction happened in Simons process using first principles methods.The process of  adsorption of CH4 and co-adsorption of CH4 and HF have been fully discussed. This work can help give a better understanding of Simons' process. 

Here are my comments. 

1. For the computational details, have you tried different functional or basis sets ? Some of the adsorption energies are very small, like in Table1 (d), which is only 0.2eV. Could you provide some data to demonstrate that the current method has achieved this level of accuracy?

2. In line 62, you mentioned your surface enhanced stability, could you provide some data to support it? How is stability of other surface like NiF3 and NiF4 ?

3. In line 160, what method did you use to calculate the vibrational frequencies? 

Reviewer 2 Report

Comments and Suggestions for Authors

This is a very nice paper using periodic DFT to model the interaction of CH4 with a NiF2 surface to develop a model of the Simons electrochemical fluorination process. They have done a nice job in mapping out the reaction coordinate for partial fluorination of CH4.

The computational method is appropriate using the PBE functional for the calculations and the NEB method to find the transition states.

A concern is the number of decimal places in eV. They give 4 decimal places, but the method is no better than 2 decimal places in eV in terms of accuracy. This need to be changed.

In figure 6, the authors need to discuss why the NEB barrier in Figure 6 is so high. This barrier is much higher than a Ni-F or C-F bond so it would have to be a dissociative process. The authors need to discuss this in more detail. Also, what does the imaginary frequency motion for the transition state look like? Does it mane sense chemically? What is being transferred, H2 or F2? I cannot tell from the plot in Figure 7.

Please provide the bond energies for Ni-F, C-F, and C-H bonds to help explain what is happening on the surface in terms of the energetics.

Please use different colors for the different atom types in the figures and give the colors in the captions.

Reviewer 3 Report

Comments and Suggestions for Authors

The manuscript "First-Principles Study of Adsorption of CH4 on a Fluorinated Model NiF2 Surface" presents a detailed investigation into the fluorination mechanisms of methane on a fluorinated nickel surface, which is relevant to understanding the Simons process used industrially for fluorination reactions. The study leverages periodic density functional theory (DFT) to explore both the adsorption of methane (CH4) and the co-adsorption of methane and hydrogen fluoride (HF) on a model NiF2 surface.

Based on the content, it is recommended that this manuscript be published in Materials, after the authors address the following questions.

1. The choice of a twice oxidized (NiF2)F2 (001) surface as the model could be better justified. Discussing why this was selected over other possible surfaces, using experimental or theoretical precedents, might clarify the model's advantages and limitations. Recognizing the limitations of the selected model surface and suggesting further studies on more complex or experimentally relevant surfaces would provide a clear direction for future research.

2. The manuscript categorizes adsorption as chemisorption or physisorption in Figure 2. Clarifying these definitions within this specific system would be helpful. Additionally, including three-dimensional isosurfaces of charge density difference for the adsorbed species might provide further clarity on the adsorption mechanisms.

3. The maximum adsorption energy reported is -10.28 eV, which appears exceptionally high. If this value is accurate, an explanation of why it is so significant would be valuable.

4. The NEB calculations presented with only four images may be insufficient. Including more images (>10) could better illustrate the energy profile, and discussing the energy barriers in relation to known activation energies might offer more detailed mechanistic insights.
